# HyperFace: Generating Synthetic Face Recognition Datasets by Exploring Face Embedding Hypersphere

**Hatef Otroshi Shahreza**[1,2] and **Sébastien Marcel**[1,3]
[1]Idiap Research Institute, Martigny, Switzerland
[2]École Polytechnique Fédérale de Lausanne (EPFL), Lausanne, Switzerland
[3]Université de Lausanne (UNIL), Lausanne, Switzerland
{hatef.otroshi,sebastien.marcel}@idiap.ch

## Abstract

Face recognition datasets are often collected by crawling Internet and without individuals' consents, raising ethical and privacy concerns. Generating synthetic datasets for training face recognition models has emerged as a promising alternative. However, the generation of synthetic datasets remains challenging as it entails adequate inter-class and intra-class variations. While advances in generative models have made it easier to increase intra-class variations in face datasets (such as pose, illumination, etc.), generating sufficient inter-class variation is still a difficult task. In this paper, we formulate the dataset generation as a packing problem on the embedding space (represented on a hypersphere) of a face recognition model and propose a new synthetic dataset generation approach, called HyperFace. We formalize our packing problem as an optimization problem and solve it with a gradient descent-based approach. Then, we use a conditional face generator model to synthesize face images from the optimized embeddings. We use our generated datasets to train face recognition models and evaluate the trained models on several benchmarking real datasets. Our experimental results show that models trained with HyperFace achieve state-of-the-art performance in training face recognition using synthetic datasets. Project page: https://www.idiap.ch/paper/hyperface

## 1 Introduction

Recent advances in the development of face recognition models are mainly driven by the deep neural networks (He et al., 2016), the angular loss functions (Deng et al., 2019; Kim et al., 2022), and the availability of large-scale training datasets (Guo et al., 2016; Cao et al., 2018; Zhu et al., 2021). The large-scale training face recognition datasets are collected by crawling the Internet and without the individual's consent, raising privacy concerns. This has created important ethical and legal challenges regarding the collecting, distribution, and use of such large-scale datasets (Nat, 2022). Considering such concerns, some popular face recognition datasets, such as MS-Celeb (Guo et al., 2016) and VGGFace2 (Cao et al., 2018), have been retracted.

With the development of generative models, generating synthetic datasets has become a promising solution to address privacy concerns in large-scale datasets (Melzi et al., 2024; Shahreza et al., 2024). In spite of several face generator models in the literature (Deng et al., 2020; Karras et al., 2019, 2020; Rombach et al., 2022; Chan et al., 2022), generating a synthetic face recognition model that can replace real face recognition datasets and be used to train a new face recognition model from scratch is a challenging task. In particular, the generated synthetic face recognition datasets require adequate *inter-class* and *intra-class* variations. While conditioning the generator models on different attributes can help increasing *intra-class* variations, increasing *inter-class* variations remains a difficult task.

In this paper, we focus on the generation of synthetic face recognition datasets and formulate the dataset generation process as a packing problem on the embedding space (represented on the surface of a hypersphere) of a pretrained face recognition model. We investigate different packing strategies

NeurIPS 2024 Safe Generative AI Workshop.

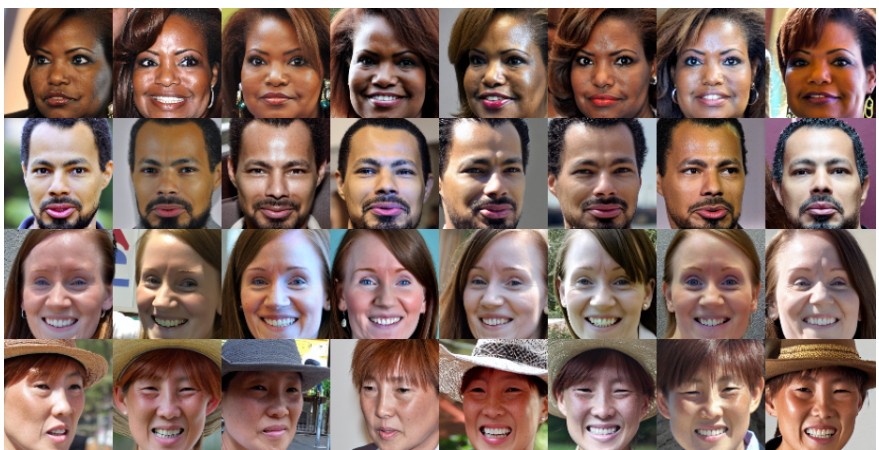

Figure 1: Sample face images from the HyperFace dataset

and show that with a simple optimization, we can find a set of reference embeddings for synthetic subjects that has a high inter-class variation. We also propose a regularization term in our optimization to keep the optimized embedding on the manifold of face embeddings. After finding optimized embeddings, we use a face generative model that can generate face images from embeddings on the hypersphere, and generate synthetic face recognition datasets. We use our generated synthetic face recognition datasets, called HyperFace, to train face recognition models. We evaluate the recognition performance of models trained using synthetic datasets, and show that our optimization and packing approach can lead to new synthetic datasets that can be used to train face recognition models. We also compare trained models with our generated dataset to models trained with previous synthetic datasets, where our generated datasets achieve competitive performance with state-of-the-art synthetic datasets in the literature. Figure 1 illustrates sample face images from our synthetic dataset.

The remainder of this paper is organized as follows. In Section 2, we present our problem formulation and describe our proposed method to generate synthetic face datasets. In Section 3, we provide our experimental results and evaluate our synthetic datasets. In Section 4, we review related work in the literature. Finally, we conclude the paper in Section 5.

## 2 Problem Formulation and Proposed Method

### 2.1 Problem Formulation

**Identity Hypersphere:** Let us assume that we have a pretrained face recognition model $F : \mathcal{I} \rightarrow \mathcal{X}$, which can extract identity features (a.k.a. embedding) $\boldsymbol{x} \in \mathcal{X} \subset \mathbb{R}_{\mathcal{X}}^n$ from each face image $\boldsymbol{I} \in \mathcal{I}$. Without loss of generality, we can assume that the extracted identity features cover a unit hypersphere[1], i.e., $||\boldsymbol{x}||_2 = 1, \forall \boldsymbol{x} \in \mathcal{X}$.

**Representing Synthetic Dataset on the Identity Hypersphere:** We can represent a synthetic face recognition dataset $\mathcal{D}$ on this hypersphere by finding the embeddings for each image in the dataset. For simplicity, let us assume that for subject $i$ in the synthetic face dataset, we can have a reference face image $\boldsymbol{I}_{\mathrm{ref},i}$ and reference embedding $\boldsymbol{x}_{\mathrm{ref},i} = F(\boldsymbol{I}_{\mathrm{ref},i})$. Note that the reference face image $\boldsymbol{I}_{\mathrm{ref},i}$ may already exist in the synthetic dataset $\mathcal{D}$, otherwise we can assume the reference embedding $\boldsymbol{x}_{\mathrm{ref},i}$ as the average of embeddings of all images for subject $i$ in the dataset $\mathcal{D}$. Therefore, the synthetic face recognition dataset $\mathcal{D}$ with $n_{\mathrm{id}}$ number of subjects can be represented as a set of reference embeddings $\{\boldsymbol{x}_{\mathrm{ref},i}\}_{i=1}^{n_{\mathrm{id}}}$.

### 2.2 HyperFace Synthetic Face Dataset

**HyperFace Optimization Problem:** By representing a synthetic dataset $\mathcal{D}$ on the identity hypersphere as a set of reference embeddings $\{\boldsymbol{x}_{\mathrm{ref},i}\}_{i=1}^{n_{\mathrm{id}}}$, we can raise the question that "*How should*

---

[1]If the identity embedding $\boldsymbol{x}$ extracted by $F(.)$ is not normalized, we normalize it such that $||\boldsymbol{x}||_2 = 1$.

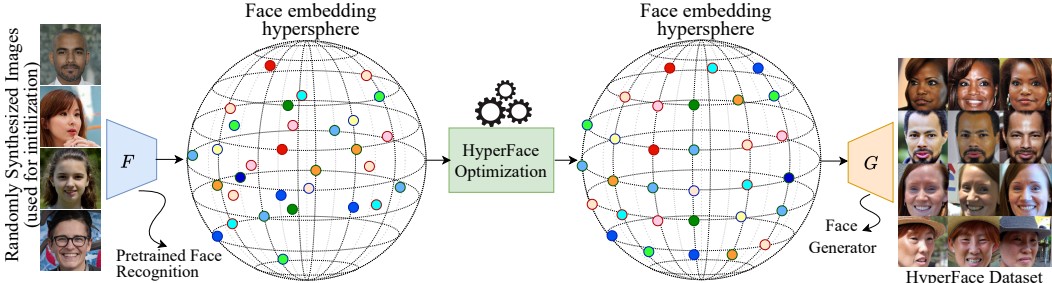

Figure 2: Block diagram of HyperFace Dataset Generation

*reference embeddings cover the identity hypersphere?*" To answer this question, we remind that the distances between reference embeddings indicate the inter-class variation in the synthetic face recognition dataset $\mathcal{D}$. Therefore, since we would like to have a high inter-class variation in the generated dataset $\mathcal{D}$, we can say that we need to maximize the distances between reference embeddings $\{\boldsymbol{x}_{\mathrm{ref},i}\}_{i=1}^{n_{\mathrm{id}}}$. In other words, we need to solve the following optimization problem:

$$\max_{\{\boldsymbol{x}_{\mathrm{ref}}\},i\neq j} \min d(\boldsymbol{x}_{\mathrm{ref},i}, \boldsymbol{x}_{\mathrm{ref},j}) \qquad \text{subject to} \quad ||\boldsymbol{x}_{\mathrm{ref},k}||_2 = 1, \forall k \in \{1, ..., n_{\mathrm{id}}\} \qquad (1)$$

where $d(\cdot, \cdot)$ is a distance function.

**Solving the HyperFace Optimization:** The optimization problem stated in Eq. 1 is a well-known optimization problem, which is known as spherical code optimization (J. H. Conway, 1998) or the Tammes problem (Tammes, 1930), where the goal is to pack a given number of points (e.g., particles, pores, electrons, etc.) on the surface of a unit sphere such that the minimum distance between points is maximized. There are different approaches for solving this optimization problem (such as geometric optimization). However, for a large dimension of hypersphere (i.e., $n_{\mathcal{X}}$) and a large number of points (i.e., number of subjects $n_{\mathrm{id}}$ in our problem), solving this optimization can be computationally expensive. To address this issue, we solve the optimization problem with an iterative approach based on gradient descent. To this end, we can randomly initialize the reference embeddings and find the optimised reference embeddings $\{\boldsymbol{x}_{\mathrm{ref},i}\}_{i=1}^{n_{\mathrm{id}}}$ using the Adam optimizer (Kingma & Ba, 2015). This allows us to solve the optimization with a reasonable computation resource. For example, we can solve the optimization for $n_{\mathcal{X}} = 512$ and $n_{\mathrm{id}} = 10,000$ on a system equipped with a single NVIDIA 3090 GPU in 6 hours.

**Regularization:** While we solve the optimization problem in Eq. 1 on the surface of a hypersphere, we should note that the manifold of embeddings $\mathcal{X}$ does not necessarily cover the whole surface of the hypersphere. This means if we get out of the distribution of embeddings $\mathcal{X}$, we may not be able to generate face images from such embeddings. Therefore, we need to add a regularization term to our optimization problem that tends to keep the reference embeddings on the manifold of embeddings $\mathcal{X}$. To this end, we consider a set of face images $\{\boldsymbol{I}_i\}_{i=1}^{n_{\mathrm{gallery}}}$ as a gallery of images[2] and extract their embeddings to have set of valid embeddings $\{\boldsymbol{x}_i\}_{i=1}^{n_{\mathrm{gallery}}}$. Then, we try to minimize the distance of our reference embeddings $\{\boldsymbol{x}_{\mathrm{ref},i}\}_{i=1}^{n_{\mathrm{id}}}$ to the set of embeddings $\{\boldsymbol{x}_i\}_{i=1}^{n_{\mathrm{gallery}}}$, which approximates the manifold of embeddings $\mathcal{X}$. To this end, for each reference embedding $\{\boldsymbol{x}_{\mathrm{ref},i}\}_{i=1}^{n_{\mathrm{id}}}$, we find the closest embedding in $\{\boldsymbol{x}_i\}_{i=1}^{n_{\mathrm{gallery}}}$ and minimize their distance. We can write the optimization in Eq. 1 as a regularized min-max optimization as follows:

$$\min_{\{\boldsymbol{x}_{\mathrm{ref}}\},i\neq j} \max -d(\boldsymbol{x}_{\mathrm{ref},i}, \boldsymbol{x}_{\mathrm{ref},j}) + \alpha \underbrace{\frac{1}{n_{\mathrm{id}}} \sum_{k=1}^{n_{\mathrm{id}}} \min_{\{\boldsymbol{x}_g\}_{g=1}^{n_{\mathrm{gallery}}}} d(\boldsymbol{x}_{\mathrm{ref},k}, \boldsymbol{x}_g)}_{\text{regularization}};$$

$$\text{subject to} \quad ||\boldsymbol{x}_{\mathrm{ref},k}||_2 = 1, \forall k \in \{1, ..., n_{\mathrm{id}}\},$$

(2)

---

[2]The gallery of face images $\{\boldsymbol{I}_i\}_{i=1}^{n_{\mathrm{gallery}}}$ can be generated using an unconditional face generator network such as StyleGAN (Karras et al., 2020), Latent Diffusion Model (LDM) (Rombach et al., 2022), etc.

---

**Algorithm 1** HyperFace Optimization for Finding Reference Embeddings

---

1: **Inputs**:    $\lambda$ : learning rate, $n_{\text{itr}}$ : number of iterations, $\{\boldsymbol{x}_g\}_{g=1}^{n_{\text{gallery}}}$ : embeddings of a gallery of face images,
2:        $\alpha$ : hyperparameter (contribution of regularization).
3: **Output**:    $\boldsymbol{X}_{\text{ref}} = \{\boldsymbol{x}_{\text{ref},i}\}_{i=1}^{n_{\text{id}}}$ : optimized reference embeddings.
4: **Procedure:**
5:     Initialize reference embeddings $\{\boldsymbol{x}_{\text{ref},i}\}_{i=1}^{n_{\text{id}}}$
6:     **For** $n = 1, .., n_{\text{itr}}$ **do**
7:       Find $\boldsymbol{x}_{\text{ref},i}, \boldsymbol{x}_{\text{ref},j} \in \boldsymbol{X}_{\text{ref}}$ which have minimum distance $d(\boldsymbol{x}_{\text{ref},i}, \boldsymbol{x}_{\text{ref},j})$
8:       $\text{Reg} \leftarrow \frac{1}{n_{\text{id}}} \sum_{k=1}^{n_{\text{id}}} \min_{\{\boldsymbol{x}_g\}_{\text{gallery}}} d(\boldsymbol{x}_{\text{ref},k}, \boldsymbol{x}_g)$              ▷ Calculate the regularization term
9:       $\text{cost} \leftarrow -d(\boldsymbol{x}_{\text{ref},i}, \boldsymbol{x}_{\text{ref},j})$
10:      $\boldsymbol{X}_{\text{ref}} \leftarrow \boldsymbol{X}_{\text{ref}} - \text{Adam}(\nabla \text{cost}, \lambda)$
11:      $\boldsymbol{X}_{\text{ref}} \leftarrow \text{normalize}(\boldsymbol{X}_{\text{ref}})$     ▷ To ensure that resulting embeddings $\boldsymbol{X}_{\text{ref}}$ remain on the hypersphere.
12:     **End For**
13: **End Procedure**

---

where $\alpha$ is a hyperparameter that controls the contribution of the regularization term in the optimization. To provide more flexibility in our optimization, we consider the size of gallery $n_{\text{gallery}}$ to be greater or equal to the number of identities $n_{\text{id}}$ in the synthetic dataset (i.e., $n_{\text{gallery}} \geq n_{\text{id}}$).

**Initialization:**    To solve the HyperFace optimization problem in Eq. 1 using Algorithm 1, we need to initialize the reference embeddings $\{\boldsymbol{x}_{\text{ref},i}\}_{i=1}^{n_{\text{id}}}$. To this end, we can generate $n_{\text{id}}$ number random synthetic images $\{\boldsymbol{I}_i\}_{i=1}^{n_{\text{id}}}$ using a face generator model, such as StyleGAN (Karras et al., 2020), Latent Diffusion Model (LDM) (Rombach et al., 2022). These models use a noise vector as the input and can generate synthetic face images in an unconditional setting. Then, after generating $\{\boldsymbol{I}_i\}_{i=1}^{n_{\text{id}}}$ images, we can extract their embeddings using the face recognition model $F(\cdot)$ and use the extracted embeddings as initialization values for the reference embeddings $\{\boldsymbol{x}_{\text{ref},i}\}_{i=1}^{n_{\text{id}}}$ in Algorithm 1.

**Image Generation:**    After we find the reference embeddings $\{\boldsymbol{x}_{\text{ref},i}\}_{i=1}^{n_{\text{id}}}$ using Algorithm 1, we can use an identity-conditioned image generator model to generate face images from reference embeddings. To this end, we use a recent face generator network (Papantoniou et al., 2024), which is based on probabilistic diffusion models. The diffusion face generator model $G(\cdot, \cdot)$ can generate a face image $\boldsymbol{I} = G(\boldsymbol{x}_{\text{ref}}, \boldsymbol{z})$ from reference embedding $\boldsymbol{x}_{\text{ref}}$ and a random noise $\boldsymbol{z} \sim \mathcal{N}(0, \mathbb{I}^{\text{DM}})$. Therefore, by changing the random noise $\boldsymbol{z}$ and sampling different noise vectors, we can generate different samples for the reference embedding $\boldsymbol{x}_{\text{ref}}$. In addition, to increase intra-class variation, we add Gaussian noise $\boldsymbol{v} \sim \mathcal{N}(0, \mathbb{I}^{n_{\mathcal{X}}})$ to the reference embedding $\boldsymbol{x}_{\text{ref}}$, and then normalize it to locate it on the hypersphere. In summary, we can generate different samples for each reference embedding $\boldsymbol{x}_{\text{ref}}$ by changing $\boldsymbol{z}$ and $\boldsymbol{v}$ noise vectors as follows:

$$\boldsymbol{I} = G(\frac{\boldsymbol{x}_{\text{ref}} + \beta \boldsymbol{v}}{||\boldsymbol{x}_{\text{ref}} + \beta \boldsymbol{v}||_2}, \boldsymbol{z}), \quad \boldsymbol{v} \sim \mathcal{N}(0, \mathbb{I}^{n_{\mathcal{X}}}), \boldsymbol{z} \sim \mathcal{N}(0, \mathbb{I}^{\text{DM}}), \tag{3}$$

where $\beta$ is a hyperparamter that controls the variations to the reference embedding. Figure 2 depicts the block diagram of our synthetic dataset generation process.

## 3 Experiments

### 3.1 Experimental Setup

**Dataset Genration:**    For solving the HyperFace optimization in Algorithm 1, we use an initial learning rate of $\lambda = 0.01$ and reduce the learning rate by power $0.75$ every $5,000$ iterations for a total number of iterations $n_{\text{itr}} = 100,000$. We also consider cosine distance, which is commonly used in face recognition systems for the comparison of face embeddings, as our distance function $d(\cdot, \cdot)$. For the hyperparameters $\alpha$ and $\beta$, we consider default values of $0.5$ and $0.01$, respectively, in our experiments. We also consider the size of gallery to be the same as the number of identities, and explore other cases where $n_{\text{gallery}} > n_{\text{id}}$ in our ablation study. We generate 64 images, by default, per each identity in our generated datasets and explore other numbers of images in our ablation study.

Table 1: Comparison of recognition performance of face recognition models trained with different synthetic datasets and a real dataset (i.e., CASIA-WebFace). The performance reported for each dataset is in terms of accuracy and best value for each benchmark is emboldened.

| Dataset name | # IDs | # Images | LFW | CPLFW | CALFW | CFP | AgeDB |
|---|---|---|---|---|---|---|---|
| SynFace (Qiu et al., 2021) | 10'000 | 999'994 | 86.57 | 65.10 | 70.08 | 66.79 | 59.13 |
| SFace (Boutros et al., 2022) | 10'572 | 1'885'877 | 93.65 | 74.90 | 80.97 | 75.36 | 70.32 |
| Syn-Multi-PIE (Colbois et al., 2021) | 10'000 | 180'000 | 78.72 | 60.22 | 61.83 | 60.84 | 54.05 |
| IDnet (Kolf et al., 2023) | 10'577 | 1'057'200 | 84.48 | 68.12 | 71.42 | 68.93 | 62.63 |
| ExFaceGAN (Boutros et al., 2023b) | 10'000 | 599'944 | 85.98 | 66.97 | 70.00 | 66.96 | 57.37 |
| GANDiffFace (Melzi et al., 2023) | 10'080 | 543'893 | 94.35 | 76.15 | 79.90 | 78.99 | 69.82 |
| Langevin-Dispersion (Geissbühler et al., 2024) | 10'000 | 650'000 | 94.38 | 65.75 | 86.03 | 65.51 | 77.30 |
| Langevin-DisCo (Geissbühler et al., 2024) | 10'000 | 650'000 | 97.07 | 76.73 | 89.05 | 79.56 | 83.38 |
| DigiFace-1M (Bae et al., 2023) | 109'999 | 1'219'995 | 90.68 | 72.55 | 73.75 | 79.43 | 68.43 |
| IDiff-Face (Uniform) (Boutros et al., 2023a) | 10'049 | 502'450 | 98.18 | 80.87 | 90.82 | 82.96 | 85.50 |
| IDiff-Face (Two-Stage) (Boutros et al., 2023a) | 10'050 | 502'500 | 98.00 | 77.77 | 88.55 | 82.57 | 82.35 |
| DCFace (Kim et al., 2023) | 10'000 | 500'000 | 98.35 | 83.12 | **91.70** | 88.43 | **89.50** |
| **HyperFace [ours]** | 10'000 | 640'000 | **98.67** | **84.68** | 89.82 | **89.14** | 87.07 |
| CASIA-WebFace (Yi et al., 2014) | 10'572 | 490'623 | 99.42 | 90.02 | 93.43 | 94.97 | 94.32 |

We use ArcFace (Deng et al., 2019) as the pretrained face recognition model $F(\cdot)$ with the embedding dimension of $n_{\mathcal{X}} = 512$ and use a pretrained generator model (Papantoniou et al., 2024) to generate face images from ArcFace embeddings. After generating face images, we align all face images using a pretrained MTCNN (Zhang et al., 2016) face detector model. For our regularization, we randomly generate images with StyleGAN (Karras et al., 2020) as default, and investigate other generator models in our ablation study.

**Evaluation:** To evaluate the generated synthetic datasets, we use each generated datasets as a training dataset for training a face recognition model. To this end, we use the iResNet50 backbone and train the model with AdaFace loss function (Kim et al., 2022) using the Stochastic Gradient Descent (SGD) optimizer with the initial learning rate 0.1 and a weight decay of $5 \times 10^{-4}$ for 30 epochs with the batch size of 256. After training the face recognition model with the synthetic dataset, we benchmark the performance of the trained face recognition models on different benchmarking datasets of real images, including Labeled Faces in the Wild (LFW) (Huang et al., 2008), Cross-age LFW (CA-LFW) (Zheng et al., 2017), CrossPose LFW (CP-LFW) (Zheng & Deng, 2018), Celebrities in Frontal-Profile in the Wild (CFP-FP) (Sengupta et al., 2016), and AgeDB-30 (Moschoglou et al., 2017) datasets. For consistency with prior works, we report recognition accuracy calculated using 10-fold cross-validation for each of benchmarking datasets. The source code of our experiments and generated datasets are publicly available[3].

## 3.2 Analysis

**Comparison with Previous Synthetic Datasets:** We compare the recognition performance of face recognition models trained with our synthetic dataset and previous synthetic datasets in the literature. We use the published dataset for each method and train all models with the same configuration for different datasets to prevent the effect of other hyperparameters (such as number of epochs, batch size, etc.). For a fair comparison, we consider the versions of datasets with a similar number of identities[4], if there are different datasets available for each method. Table 1 compares the recognition performance of face recognition models trained with different synthetic datasets. As the results in this table show, our method achieves state-of-the-art performance in training face recognition using synthetic data. Figure 1 illustrates sample face images from our synthetic dataset. Figure 3 of appendix also presents more sample images from HyperFace dataset.

---

[3]Project page: https://www.idiap.ch/paper/hyperface

[4]Only in the dataset used for DigiFace (Bae et al., 2023) there are more identities, because there is only one version available for this dataset, which has a greater number of identities compared to other existing synthetic datasets.

**Ablation Study:** In our dataset generation method, there are different hyperparameters which can affect the HyperFace optimization and the generated synthetic datasets. Table 2 reports the ablation study on the number of images generated per each synthetic identity in our experiments. As the results in Table 2 show, increasing the number of images per identity improves the recognition performance of trained face recognition model, but it tends to saturate after 64 images per identity.

Table 3 also compares the number of identities in the generated dataset. As the results in Table 3 show, increasing the number of identities improves the recognition performance of trained face recognition model on the benchmarking datasets.

Table 4 reports the recognition performance achieved for face recognition model trained with datasets optimized with different number images. as the results in this table shows, increasing the size of gallery improves the performance of the trained model. However, with 10,000 images we can still approximate the manifold of face embeddings on the hypersphere.

As another ablation study, we use different source of images for the gallery set to use in our regularization and solve the HyperFace optimization. We use pretrained StyleGAN (Karras et al., 2020) as a GAN-based generator model and a pretrained latent diffusion model (Rombach et al., 2022) as a diffusion-based generator model. We use these generator models and randomly generate some synthetic face images. In addition, for our ablation study, we consider some real images from BUPT dataset (Wang et al., 2019) as a dataset of real face images. As the results in Table 5 show, optimization with images from StyleGAN and LDM lead to comparable performance for the generated face recognition dataset. However, the real images in the BUPT dataset lead to superior performance. This suggests that the synthesized images cannot completely cover the manifold of embeddings and if we use real images as our gallery it can improve the generated dataset and recognition performance of our face recognition model.

We also study the effect of hyperparameters $\alpha$ and $\beta$ on the generated face recognition dataset. Table 6 reports the ablation study for the contribution of regularization in our optimization ($\alpha$). As the results in this table shows, the regularization enhances the quality of generated dataset and improves the recognition performance of face recognition model.

Table 2: Ablation study on the effect of number of images

| Image/ID | LFW | CPLFW | CALFW | CFP | AgeDB |
|---|---|---|---|---|---|
| 32 | **98.70** | 84.17 | 88.83 | 88.74 | 86.33 |
| 50 | 98.50 | 84.23 | 89.40 | 88.83 | 86.53 |
| 64 | 98.67 | **84.68** | 89.82 | 89.14 | 87.07 |
| 96 | 98.42 | 84.15 | 89.00 | **89.51** | 87.45 |
| 128 | 98.20 | 83.63 | **89.82** | 89.31 | **87.62** |

Table 3: Ablation study on the effect of number of identities

| # IDs | LFW | CPLFW | CALFW | CFP | AgeDB |
|---|---|---|---|---|---|
| 10k | 98.67 | 84.68 | 89.82 | 89.14 | 87.07 |
| 30k | **98.82** | **85.23** | **91.12** | **91.74** | **89.42** |

Table 4: Ablation study on the effect of $n_{\text{gallery}}$

| $n_{\text{gallery}}$ | LFW | CPLFW | CALFW | CFP | AgeDB |
|---|---|---|---|---|---|
| 10K | 98.53 | 84.00 | 88.92 | **89.34** | 85.9 |
| 20K | 98.50 | **84.32** | **89.28** | 89.17 | 86.00 |
| 50K | **98.72** | 84.23 | 88.72 | 89.19 | **86.85** |

Table 5: Ablation study on the type of data in gallery

| Gallery | LFW | CPLFW | CALFW | CFP | AgeDB |
|---|---|---|---|---|---|
| StyleGAN | 98.67 | 84.68 | 89.82 | 89.14 | 87.07 |
| LDM | 98.65 | 84.35 | 89.17 | 89.09 | 86.35 |
| BUPT | **98.70** | **84.77** | **90.03** | 89.16 | **87.13** |

Table 6: Ablation study on the effect of $\alpha$

| $\alpha$ | LFW | CPLFW | CALFW | CFP | AgeDB |
|---|---|---|---|---|---|
| 0 | 98.40 | 84.15 | 88.87 | 89.31 | 86.48 |
| 0.50 | **98.67** | 84.68 | **89.82** | 89.14 | **87.07** |
| 0.75 | 98.62 | 84.32 | 89.48 | 89.67 | 86.72 |
| 1.0 | 98.55 | **84.72** | 89.1 | **89.76** | 86.63 |

Similarly, Table 7 reports the ablation study for the effect of noise in data generation and augmentation (i.e., hyperparamter $\beta$ in in Eq. 3). As can be seen, the added noise increases the variation for images of each subject and increases the performance of face recognition models trained with the generated datasets.

Table 7: Ablation study on the effect of $\beta$

| $\beta$ | LFW | CPLFW | CALFW | CFP | AgeDB |
|---|---|---|---|---|---|
| 0 | 98.53 | 84.00 | 88.92 | 89.34 | 85.9 |
| 0.005 | 98.67 | 84.68 | 89.82 | 89.14 | 87.07 |
| 0.010 | **98.7** | **84.72** | 90.05 | 89.54 | 88.42 |
| 0.020 | 98.4 | 84.05 | **91.32** | **90.13** | **89.83** |

As another experiment, we consider different backbones and train face recognition models with different number of layers. As the results in Table 8 show, increasing the number of layers improve the recognition performance of trained face recognition model. While this is expected and has

Table 8: Ablation study on the network structure

| Network | LFW | CPLFW | CALFW | CFP | AgeDB |
|---|---|---|---|---|---|
| ResNet18 | 98.33 | 81.38 | 88.53 | 86.03 | 85.27 |
| ResNet34 | 98.5 | 83.47 | 88.88 | 88.29 | 86.42 |
| ResNet50 | 98.67 | 84.68 | 89.82 | 89.14 | 87.07 |
| ResNet101 | **98.73** | **85.43** | **90.05** | **89.54** | **87.52** |

been observed for training using large-scale face recognition datasets, it sheds light on more potentials in the generated synthetic datasets.

**Scaling Dataset Generation:** To increase the size of the synthetic face recognition dataset, we can increase the number of images per identity and also the number of samples per identity. In our ablation study, we investigated the effect of the number of images (Table 2) and the number of identities (Table 3) on the recognition performance of the face recognition model. However, increasing the size of the dataset requires more computation. Increasing the number of images in the dataset has linear complexity in our image generation step (i.e., $\mathcal{O}(n_{\text{images}})$, where $n_{\text{images}}$ is the number of images in the generated dataset). However, the complexity of solving the HyperFace optimization problem with iterative optimization in Algorithm 1 has quadratic complexity (i.e., $\mathcal{O}(n_{\text{id}}^2)$). Therefore, solving this optimization for a larger number of identities requires much more computation resources. Meanwhile, most existing synthetic datasets in the literature have a comparable number of identities to our experiments. We should note that in our optimization, we considered all points in each iteration of optimization which introduces quadratic complexity to our optimization. However, we can solve the optimization with stochastic mini-batches of points on the embedding hypersphere, which can reduce the complexity in each iteration (i.e., $\mathcal{O}(b^2)$, where $b$ is batch size and $b \leq n_{\text{id}}$), but may increase the optimization error.

## 4 Related Work

With the advances in generative models, several synthetic face recognition datasets have been proposed in the literature. Bae et al. (2023) proposed DigiFace dataset where they used a computer-graphic pipeline to render different identities and also generate different images for each identity by introducing different variations based on face attributes (e.g., variation in facial pose, accessories, and textures). In contrast to (Bae et al., 2023) , other papers in the literature used Generative Adversarial Networks (GANs) or probabilistic Diffusion Models (PDMs) to generate synthetic face datasets. Qiu et al. (2021) proposed SynFace and utilised DiscoFaceGAN (Deng et al., 2020) to generate their dataset. They generated different synthetic identities using identity mixup by exploring the latent space of DiscoFaceGAN to increase intra-class variation and then used DiscoFaceGAN to generate different images for each identity.

Boutros et al. (2022) proposed SFace by training an identity-conditioned StyleGAN (Karras et al., 2020) on the CASIA-WebFace (Yi et al., 2014) and then generating the SFace dataset using the trained model. Kolf et al. (2023) also trained an identity-conditioned StyleGAN (Karras et al., 2020) in a three-player GAN framework to integrate the identity information into the generation process and proposed the IDnet dataset. Colbois et al. (2021) proposed the Syn-Multi-PIE dataset using a pretrained StyleGAN (Karras et al., 2020). They trained a support vector machine (SVM) to find directions for different variations (such as pose, illuminations, etc.) in the intermediate latent space of a pretrained StyleGAN. Then, they used StyleGAN to generate different identities and synthesized different images for each identity by exploring the intermediate latent space of StyleGAN

using linear combinations of calculated directions. Boutros et al. (2023b) proposed ExFaceGAN, where they used SVM to disentangle the identity information in the latent space of pretrained GANs, and then generated different identities with several images within the corresponding identity boundaries. Geissbühler et al. (2024) used stochastic Brownian forces to sample different identities in the intermediate latent space of pretrained StyleGAN (Karras et al., 2020) and generate different identities (named Langavien). Then they solved a similar dynamical equation in the latent space of StyleGAN to generate different images for each identity (named Langavien-Dispersion) and further explored the intermediate latent space of StyleGAN (named Langavien-DisCo).

Melzi et al. (2023) proposed GANDiffFace, a hybrid dataset generation framework, where they used StyleGAN to generate face images with different identities, and then used DreamBooth (Ruiz et al., 2023) as a diffusion-based generator, to generate different samples for each identity. Boutros et al. (2023a) trained an identity-conditioned diffusion model to generate synthetic face images and proposed IDiffFace datasets. They generated different samples using an unconditional model, and then generated different samples using their conditional diffusion model (named IDiff-Face Two-Stage). Alternatively, they uniformly sampled different identities and generated different samples for each identity using their identity-conditioned diffusion model (named IDiff-Face Uniform). Kim et al. (2023) proposed DCFace, where they trained a dual condition (style and identity conditions) face generator model based on diffusion models on the CASIA-WebFace dataset. They used their trained diffusion model to generate different identities and different styles for each identity by varying identity and style conditions.

## 5 Conclusion

In this paper, we formalized the dataset generation as a packing problem on the hypersphere of a pretrained face recognition model. We focused on inter-class variation and designed our packing problem to increase the distance between synthetic identities. Then, we considered our packing problem as a regularized optimization and solved it with an iterative gradient-descent-based approach. Since the manifold of face embeddings does not cover the whole hypersphere, the regularization allows us to approximate the manifold of face embeddings and enhance the quality of generated face images. We used the generated datasets by our method (called HyperFace) to train face recognition models, and evaluated the trained models on several real benchmarking datasets. Our experiments demonstrate the effectiveness of our approach, which achieves state-of-the-art performance for training face recognition using synthetic data. We also presented an extensive ablation study to investigate the effect of each hyperparameter in our dataset generation method.

## Acknowledgments
This research is based upon work supported by the Hasler foundation through the "Responsible Face Recognition" (SAFER) project.

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

# A    Sample Images from HyperFace Dataset

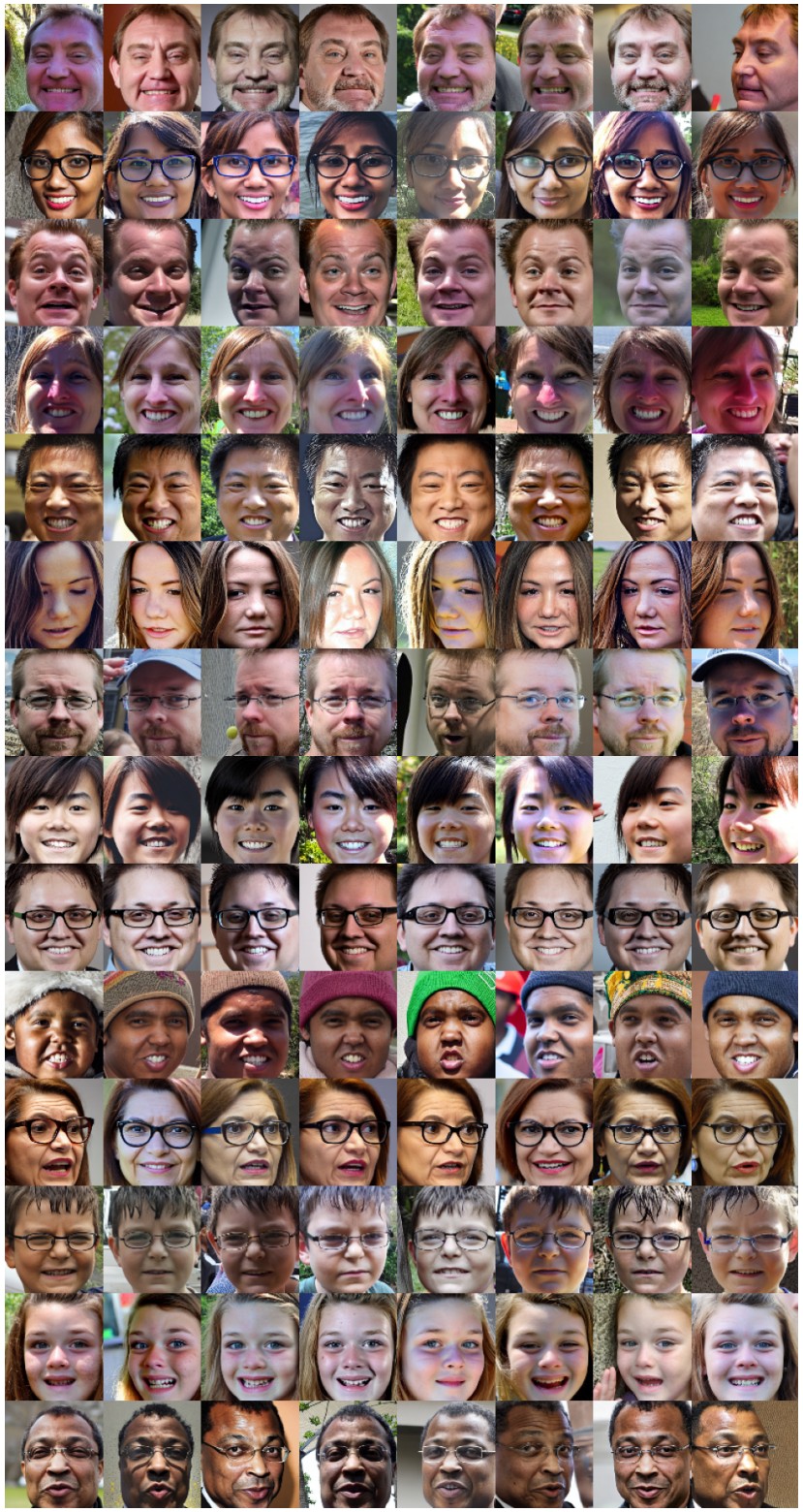

Figure 3: Sample face images from the HyperFace dataset

# B  Leakage of Identity

In our dataset generation method, we used images synthesized by StyleGAN for initialization and regularization. Therefore, it is important if there is any leakage of privacy data in the images generated from StyleGAN in the final generated dataset. To this end, similar to (Shahreza & Marcel, 2024), we extract and compare embeddings from all the generated images to embeddings of all face images in the training dataset of StyleGAN. The highest similarity score between generated images and training dataset correspond to children images (as shown in Figure 4a) which are difficult to compare visually and conclude potential leakage. Figure 4b illustrates images of highest scores excluding children. While there are some visual similarities in the images, it is difficult to conclude leakage of identity in the generated synthetic dataset.

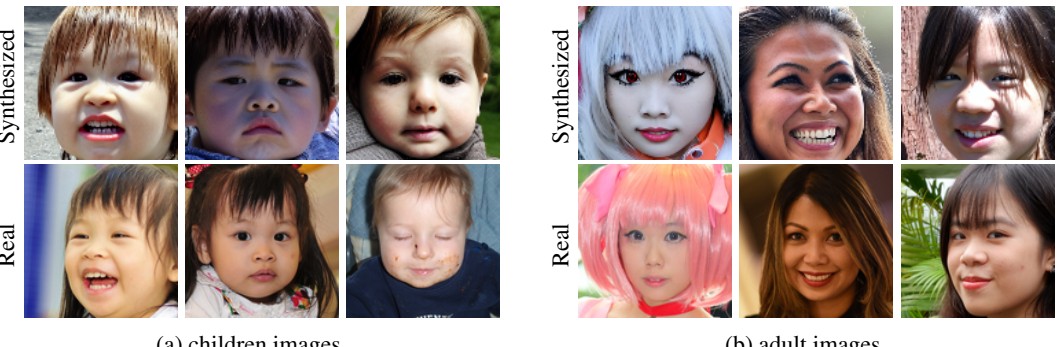

(a) children images                          (b) adult images

Figure 4: Sample pairs of images with the highest similarity between face embeddings of images in synthesized dataset and training dataset of StyleGAN, which was used to generate random images for initialization and regularization in the HyperFace optimization.

# C  Ethical Considerations

State-of-the-art face recognition models are trained with large-scale face recognition datasets, which are crawled from the Internet, raising ethical and privacy concerns. To address the ethical and privacy concerns with web-crawled data, we can use synthetic data to train face recognition models. However, generating synthetic face recognition datasets also requires face generator models which are trained from a set of real face images. Therefore, we still rely on real face images in the generation pipeline.

In our experiments, we investigated if we have leakage of identity in the generated synthetic dataset based on images used for initialization and regularization. However, there are other privacy-sensitive components used in our method. For example, we defined and solved our optimization problem on the embedding hypersphere of a pretrained face recognition model. Therefore, for generating fully privacy-friendly datasets, the leakage of information by other components needs to be investigated.

We should also note that while we tried to increase the inter-class variations in our method, there might be still a potential lack of diversity in different demography groups, stemming from implicit biases of the datasets used for training in our pipeline (such as the pretrained face recognition model, the gallery of images used for regularization, etc.). It is also noteworthy that the project on which the work has been conducted has passed an Institutional Ethical Review Board (IRB).

