# OpenReview forum: "HyperFace: Generating Synthetic Face Recognition Datasets by Exploring Face Embedding Hypersphere"
_NeurIPS.cc/2024/Workshop/SafeGenAi — SafeGenAi Poster_

### Official Review · Reviewer_F88N · 2024-10-09
**The paper makes a strong case for generating synthetic face recognition datasets using the hypersphere-based optimization approach**

**Rating:** 7
**Confidence:** 4

**Review:**

**Evaluation of Paper**

**Quality** - The paper presents a very methodical approach to synthetic face recognition dataset creation using the hypersphere concept reinforced by the comparisons with state-of-the-art datasets and ablation studies.

**Originality** - The paper's ideation of synthetic face dataset generation as a packing problem on the embedding space along with regularization term brings in a very fresh perspective while attempting to maintain data conformity.

**Clarity** - The paper presents the idea/formulated solution in systematic manner beginning from the problem formulation, methodology, experimental results and with the help of illustrations where applicable. Certain sections with mathematical equations used in coining the regularization term could be made more comprehensive.

**Significance** - The paper proposes an approach that is practical and is a prospective candidate for real world applications while addressing ethical and privacy concerns.

**Pros:**
1. The approach to synthetic face dataset generation as a hypersphere packing problem is distinctive.
2. The paper attempts to solve the ethical and privacy concerns in models trained on real world data.
3. The paper also demonstrates strong experimental validation and benchmarked synthetic face dataset generation with desired accuracy.

**Cons:**
1. Models used in experimental evaluations were pretrained and should account for the bias from the originaly training data.
2. The paper does not take into consideration bias related to demography in the synthetic face dataset created.

---

### Official Review · Reviewer_EoXz · 2024-10-09
**Review of - HyperFace: Generating Synthetic Face Recognition Datasets by Exploring Face Embedding Hypersphere**

**Rating:** 8
**Confidence:** 3

**Review:**

**Review Highlights**

This paper is throughly addresses a relevant problem with a well defined approach; demonstrates clear results. It also goes about adding detail about the comprehensive experiments, ablation studies that substantiate the method's effectiveness. In my opinion, the paper tries to tackle an vital issue in the face recognition community, addressing privacy concerns, which makes this paper valuable from an ethical standpoint. The authors could have explored a bit more into the potential biases in the synthetic datasets - this can be done as part of future exploration. Overall, this paper can be interesting to both researchers and practitioners, justifying a solid acceptance.